# Colorimetric Films Based on Polyvinyl Alcohol and Anthocyanins Extracted from Purple Tomato to Monitor the Freshness of Shrimp

**DOI:** 10.3390/polym16040495

**Published:** 2024-02-10

**Authors:** Yangyang Qi, Yana Li

**Affiliations:** School of Mechanical Engineering, Wuhan Polytechnic University, Wuhan 430023, China; qi13461805631@163.com

**Keywords:** purple tomato anthocyanin, polyvinyl alcohol, intelligent packaging, shrimp

## Abstract

Anthocyanin extracts from purple tomato (PTA) were incorporated with polyvinyl alcohol (PVA), resulting in a series of colorimetric PVA/PTA films with PTA concentrations of 0%, 1%, 3%, and 5% (based on PVA). The role of anthocyanin on color response, Fourier-transform infrared (FTIR), thickness, water content, mechanical properties, antioxidant activity, and water vapor permeability (WVP) through the films was examined. In addition, its application in smart packaging to assess the freshness of shrimp was studied. It was found that the tensile strength, contact angle and WVP of PVA/PTA films increases with the addition of more PTA, while the elongation at break and water content decreased. FTIR analysis showed that there are interactions between PTA and the PVA matrix. The addition of anthocyanins caused significant improvement in the antioxidant properties of PVA films. Furthermore, the total volatile alkaline nitrogen (TVB-N), total plate count (TPC), and pH value of shrimp were monitored after 4 days of refrigeration, and the color change of the indexes was recorded. The PVA/PTA films changed color from purple to yellow-green during the storage time of 0–4 days for shrimp. This suggests that the film could be used in smart packaging as a real-time freshness indicator for shrimp.

## 1. Introduction

Recently in the field of intelligent packaging, many researchers have focused on colorimetric indicator films because they can provide a real-time information on shelf life, food safety and quality by monitoring the environmental changes around the food inside the packaging without opening the package [1]. The ideal colorimetric indicator combines natural pigments that meet food-grade standards with a stable, solid carrier substrate and achieves color change by monitoring changes in the environment [2,3]. From these substrates, a variety of polymer materials have been selected-including but not limited to starch, cellulose, chitosan, chitin, gelatin, and polyvinyl alcohol (PVA)-as have their blends, which have been used as carrier matrices [4] in which, as a biodegradable synthetic polymer with non-toxic, good film formation and mechanical properties [5], PVA is widely applied in intelligent packaging and food packaging [6,7,8,9].

Anthocyanins, being a class of compounds that occur naturally in plants, are widely used in the food industry as food colorants and antioxidants due to being rich in color and antioxidant properties. At the same time, the significant color response against pH makes anthocyanins widely used in intelligent packaging, and more and more anthocyanin-based indicator films have been developed to monitor the freshness of meat, fish, and shrimp [1,10,11,12]. Many studies show that when the environment is acidic, anthocyanins take on a red or pink color; in alkaline environments, they will appear blue or purple. This could be explained by the fact that the stability of anthocyanins varies with differences in pH. In dilute acid solutions, anthocyanins exist in the form of flavonoid cations, and double bonds are prone to extended conjugation, forming a colorless pseudo-alkaline and balanced reaction of red flavonoid cations. When the solution’s alkalinity increases, the nucleophilic reaction between the red flavonoid cations and water forms colorless carbinol [12]. The Handerson-Hassel Balch equation (Figure 1) can be used to analyze and demonstrate the reaction equilibrium of anthocyanins [13].

Zhang et al. [14] prepared corn starch/PVA (PS) films containing different anthocyanins extracted from purple sweet potato extracts (PSPE) and red cabbage extracts (RCE) for monitoring the freshness of shrimp. Compared to PS-RCE film, PS-PSPE films exhibited more vivid color changes, better mechanical properties, and lower light transmittance at lower leachabilities, indicating that the color change of the indicator can reflect shrimp spoilage. Sani et al. [15] successfully prepared a new pH-sensitive colorimetric film by incorporating red barberry anthocyanin (RBA) into a composite chitin nanofiber and methylcellulose matrix. It was shown that RBA was responsive to changes in pH and ammonia generation. Therefore, it can be used as a colorimetric indicator to identify signs of food spoilage. The effectiveness of this indicator in assessing the freshness and spoilage status of fish was verified. It was found that the mechanical, thermal, and antibacterial properties of films were better when cooperating with anthocyanins. 

On the other hand, for intelligent films, the doped anthocyanins are from red cabbage, black carrot, purple sweet potato, blueberry, black wolfberry, and so on [13], however, there are few reports of anthocyanins from purple tomatoes. The research demonstrated that the origin of anthocyanins significantly impacts the functionality and physical properties of the indicator films [16]. In this study, the anthocyanins extracted from purple tomatoes (PTA) were used in combination with PVA to prepare colorimetric indicator films. The physicochemical properties of the PVA/PTA film were investigated, and its color sensitivity to the freshness of shrimp was examined to assess its potential as a visual indicator.

## 2. Materials and Methods

### 2.1. Materials

PVA was purchased from Shanpu Chemical Co., Ltd. (Shanghai, China) (CAS: 9002-89-5). PTA was from Zhejiang Bison Biotechnology., Ltd. (Jiaxing, China). These substances were extracted from purple tomato fruits using an efficient purification method, resulting in a purity of 54% (*w*/*w*) [12]. Glacial acetic acid (analytical purity) was purchased from Bohuatong Chemical Products Sales Center (Tianjin, China) (CAS: 64-19-7). 2,2-diphenyl-1-trinitrophenylhydrazine (DPPH) containing 95% free radicals was purchased from Sinopharm Chemical Reagent Co., Ltd. (Shanghai, China) (CAS: 1898-66-4). The shrimp were purchased from Zhongbai Warehouse in Wuhan, China. All chemicals used were high-purity analytical grade chemicals. The solvent in all formulations was deionized water.

### 2.2. Preparation of PVA/PTA Films

PVA/PTA films were prepared via the casting method. In total, 10 g of PVA was dissolved in 500 mL of deionized water and stirred vigorously to give a PVA solution. A certain amount of PTA (0.1 g, 0.3 g, or 0.5 g) was then added to the PVA solution to prepare the PVA/PTA solution. The mixture was stirred continuously for 30 min until it reached a consistency suitable for the PVA/PTA mixture. Finally, pour 80 mL of mixed PVA/PTA solution or pure PVA solution into a Petri dish with a diameter of 15 cm and dry at 45 °C for 18 h. The PVA/PTA films produced are stored in a glass dryer. The films with three different PTA concentrations are 1%, 3%, and 5% (*w*/*w*, based on dry PVA).

### 2.3. Characterization of PTA or PVA/PTA Films

#### 2.3.1. Color and UV-Vis Spectra of PTA

At varying pH values (pH 2–12), the camera was used to capture the color changes of PTA solution, and the visible light spectrum was scanned by UV-vis spectrophotometer (756PC, Shanghai Haoyu Hengping Scientific Instruments Co., Ltd., Shanghai, China) with the wavelength ranging from 200 nm to 800 nm [17].

#### 2.3.2. pH-Response of PVA/PTA Films

The prepared PVA/PTA films (4 cm × 4 cm) were exposed to buffer solutions of pH 2, 3, 4, 5, 6, 7, 8, 9, 10, 11, or 12. The films were immersed in the pH solution for 60 s and then photographed. At the same time, the *L**, *a**, and *b** of the films as a function of pH was determined using a colorimeter (CR-10, Konica Minolta, Japan). The value of the total color difference (Δ*E*) was calculated according to Equation (1).
(1)$$\Delta E=\sqrt{{\left(L^{*}-L_{0}\right)}^{2}+{\left(a^{*}-a_{0}\right)}^{2}+{\left(b^{*}-b_{0}\right)}^{2}}$$
where *L**, *a**, and *b** are the values of the samples; $L_{0}$, $a_{0}$, and $b_{0}$ represent the original values of the standard white plate, and the values of $L_{0}$, $a_{0}$, and $b_{0}$ were 92.9, 0.32, and 0.33, respectively.

#### 2.3.3. Thickness of PVA/PTA Films

The thickness measurement was performed with a digital micrometer (SM-114, Shanghai Liuling Instrument Factory, Shanghai, China). The final value is the average of 10 randomly selected points measured on the film.

#### 2.3.4. Fourier Transform Infrared (FTIR) Spectroscopy of PVA/PTA Films

The FTIR spectra of the PVA/PTA film with 5% PTA concentration and pure PVA film was obtained between wavenumbers 400 and 4000 cm^−1^ via a FTIR spectrophotometer (Tensor 27, Bruker Optics, Bremen, Germany). 

#### 2.3.5. Mechanical Properties of PVA/PTA Films

The mechanical properties were measured using an electromechanical universal testing machine (UTM4104, SUNS, Shenzhen, China) according to the standard method specified in ISO 1184-1983. The test was performed at 26 °C at a crosshead speed of 50 mm/min. The sample used was a rectangular film with a size of 100 mm × 15 mm. A total of 6 samples were tested, and the average value was taken as the final test data. All the samples were conditioned at room temperature and 50% RH for at least 48 h before testing.

#### 2.3.6. Moisture Content of PVA/PTA Films

The films were dried at 105 °C until the weight was constant and to calculate the moisture content (MC). The moisture content was calculated according to Equation (2).
(2)$$MC\left(\%\right)=\frac{M_{0}-M_{1}}{M_{0}}\times 100\%$$
where *M*_0_ represents the initial weight of the film and *M*_1_ represents the weight of the film after drying.

#### 2.3.7. Water Vapor Permeability of PVA/PTA Films

The water vapor permeability (WVP) of the samples was determined based on ASTM E96-1980 by using a moisture permeation meter (W3/031X, Labthink, Jinan, China). The film samples were cut into 74 mm diameter circles, placed in triplicate in a permeation cell containing deionized water (90% relative humidity at 28 °C), weighed, placed in a desiccator, and averaged using a water vapor transmission rate tester.

#### 2.3.8. Antioxidant Activity of PVA/PTA Films

The antioxidant activity of films was measured according to the reference [18]. The films were immersed in 95% ethanol at 65 °C for 5 h, 1 mL of the sample was blended with 4 mL of 0.1 mM DPPH dissolved in ethanol. Next, the absorbance of the mixture at a wavelength of 517 nm was determined. Each sample was tested in triplicate copies; absolute ethanol was used as the blank. The antioxidant clearance was calculated according to Equation (3).
(3)$$S\left(\%\right)=\frac{A_{0}-A_{S}}{A_{0}}$$
where $A_{S}$ and $A_{0}$ represent the absorbance values of the dipstick-treated and non-dipstick solutions, respectively.

### 2.4. Monitoring Freshness of Shrimp

#### 2.4.1. Packaging

The shrimp sample (20 g) was placed into a Petri dish and sealed by a Petri dish lid with a colorimetric indicator film (4 cm × 4 cm) that adheres to the inside of the lid [19]. Then, the samples were stored at 4 °C and 60% RH for 4 d. During the storage, the color changes of the film were photographed each day.

#### 2.4.2. Microbial Colony Measurement

The microbial colony was measured using the plate-counting method [20] to obtain the total plate count (TPC). The shrimp was transferred into bags containing 135 mL of 0.1% (*w*/*w*) saline solution and homogenized using a homogenizer (JX-05, TUOHE Electromechanical Technology, Shanghai, China) for 3 min. Then, the diluted homogenized solutions were plated on agar prior to being incubated at 37 °C for 48 h.

#### 2.4.3. pH and Total Volatile Basic Nitrogen (TVB-N) Measurement

A 20 g shrimp sample was first homogenized by mixing with 180 mL of distilled water, and then a digital pH meter (PhS-3E, INESA Scientific Instruments Co., Ltd., Shanghai, China) was used to accurately measure the pH of the resulting mixture slurry. Simultaneously, the total volatile basic nitrogen (TVB-N) in shrimp was determined by using a semi-micro Kjeldahl distiller according to the method described in Ref. [21].

### 2.5. Statistical Analysis

Data results were reported as mean ± standard deviation and one-way ANOVA was performed using SPSS software (22.0, SPSS Statistical software, Inc., Chicago, IL, USA) and using Origin software (22.0, OriginLab Corporation, Northampton, MA, USA) to perform Pearson correlation analysis. The significance difference between the means of each group was determined by Duncan’s multiple range test, and the significance level was set to *p* < 0.05. 

## 3. Results and Discussion

### 3.1. Color and UV-Vis Spectra of PTA

The visual color and UV-vis spectrum of PTA are shown in Figure 2a. PTA is sensitive to pH, changing color from purplish-red (pH 2) to pink (pH 3–4) and light purple (pH 5–7), then light blue (pH 8), and finally yellow at pH > 9 due to degradation caused by strong bases [18]. Similar to our study, Rawdkuen et al. [22] found that anthocyanins can form a stable red color (flavonoid cation) under strong acidic conditions (pH < 4). However, in the pH range of 5 to 6, they were almost colorless and could form chalcones and methanol. When the pH value was greater than 8, it formed pale yellow or colorless chalcone. Figure 2a also showed that in the pH range of 2–8, the maximum absorption peak of anthocyanins gradually shifted from 527 nm to 538 nm, and the absorption intensity of PTA decreased slowly with the increase in pH. When the pH value exceeds 9, the absorption intensity gradually increases with further increases in pH, and the maximum absorption peak shifts from 570 nm to 660 nm. UV-Vis spectroscopy also confirmed the pH sensitivity of PTA. As the pH of the solution changes, the anthocyanins undergo structural changes, resulting in a change in the absorption properties.

### 3.2. Original Films 

Digital photos and color parameters of PVA/PTA films are displayed in Figure 2b and Table 1. The color and opacity values were affected by PTA concentration. As shown in Figure 2b, the pure PVA was transparent, the films were lavender at a concentration of 1%, the films were purple at a concentration of 3%, and the films were dark purple at a concentration of 5%. These changes in color and opacity values are due to changes in PTA concentrations. From Table 1, it can be seen that the *L**, *a**, and *b** values of the films were dramatically influenced by the concentrations of anthocyanins (*p* < 0.05). With the addition of PTA, the color of the composite films has obviously changed, and the Δ*E* value increased significantly (*p* < 0.05), reaching the highest at 65.94 for the films with 5%-PTA. At the same time, the brightness (*L**) of the films decreased significantly (*p* < 0.05), and the *L** value of the 5%-PTA film was the lowest and gradually tended to black. The *a** value increased significantly (*p* < 0.05) and appeared red. The *b** value of the film was <0, indicating that the film was blue; the final result was that the observed color was a deep purple. 

### 3.3. pH-Response of Films

The films containing PTA extract exhibited a color change with pH, as shown in Figure 2c, being similar to that observed for PTA extract (Figure 2a). The color of the films shifted roughly as follows: red → colorless → blue–green → light yellow. Under strong acidic conditions (pH < 4), the reduction in the double bond in anthocyanin—i.e., the protonation of flavonoid cations-explained the observed color change. Between pH 4–10, the films were almost colorless or gray. When pH > 10, the trend was yellow-green due to the degradation of anthocyanins in strongly alkaline media. Several studies have demonstrated that anthocyanin-rich films can respond significantly to changes in pH [15].

In addition, pigment migration was observed for all the films. Increasing the pigment content, significantly more monomeric anthocyanin migrated into the media, so that at pH 2, the media was pink color for 3% PTA and 5% PTA films, and at pH 12, the solution around the 5% PTA film was yellow in color compared with the colorless solutions for 0% PTA and 3% PTA films. Due to the pigment-film interactions that developed, the polymeric chains presumably had higher relaxation ability, which led to higher swelling and thus enhanced extract release [23].

The change in ΔE with pH is also an important point when evaluating a smart indication labeling system. It is necessary to make the change in ΔE indicator visible enough so that the consumer can easily see this color change and know the quality of the product. Based on the results in Figure 2d, we can see that different films have different responses to pH, and the ΔE strongly dependents on PTA amount. It was found that the variations in ΔE for PVA/1%-PTA, PVA/3%-PTA and PVA/5%-PTA was in the ranges of 25–55, 55–60, and 65–75, respectively, in pH 2–12. This indicated that the most noticeable color change was PVA/1%-PTA, then PVA/5%-PTA, and PVA/3%-PTA was the last. From the color change results, the PVA films combined with 1% and 5% concentration of PTA have great potential as a chromogenic material in food packaging.

### 3.4. Mechanical Properties of Films

As shown in Figure 3a, the tensile strength (TS) tended to increase and the elongation at break (EAB) decreased significantly (*p* < 0.05) with the elevation of PTA concentration, showing that the addition of PTA extract could increase the mechanical resistance and decrease the flexibility of the film. The formation of hydrogen bonds between PTA and film matrix increases the intermolecular forces and tighter structure, which can increase their mechanical resistance [24]. As the thickness of the film increases with the concentration of PTA, their tensile strength is correspondingly increased. On the other hand, the addition of PTA reduces the interaction of water molecules with the film-forming matrix because water molecules can act as plasticizers to become more elastic and flexible in the film structure [25], but the increase in film thickness increases the elongation at break of the film [26]. According to the experimental results, it can be concluded that the effect of PTA on the elongation at break is greater than that of the film thickness.

In addition, in the tensile test, there was no water loss due to the low moisture content of the film, and the sample did not change color. This indicates that the moisture content of the film has no significant effect on the results of the tensile test. The low moisture content of the film may help to maintain the stability of its physical properties, thus avoiding the effect of moisture on mechanical properties.

### 3.5. Film Thickness, Moisture Content and Water Vapor Permeability of Films

The film thickness was impacted by the composition of the components and the intermolecular forces [27]. Table 2 illustrates that significant statistical differences (*p* < 0.05) were present between the films of varying concentrations. With the increase in anthocyanin concentration, the thicknesses all increased significantly by 7.5%, 18.1%, and 20.1%, respectively. The thickness of the films was influenced by the film-forming matrix and may be related to the increase in the solid content of the polymer [28]. In addition, the addition of PTA can also lead to more porosity and defects in the film, which may also be one of the reasons for the increase in thickness. Before testing the moisture content of the film, PTA was stored in a dry environment due to its hydrophilic nature. However, despite the hydrophilic nature of PTA, the phenolic hydroxyl group forms more hydrogen bonds with the hydroxyl group of the film-forming matrix, limiting the interaction of the PVA/PTA composite films with water, resulting in a decrease in MC [29]. Therefore, although the concentration of anthocyanins affects the thickness of PVA/PTA composite films, it has no obvious effect on its moisture content.

WVP is an important index to evaluate the waterproof performance of food packaging film. As shown in Table 2, there are significant differences between the 1%, 3%, and 5% concentrations of PVA/PTA films (*p* < 0.05). The WVP of the control group with the highest PTA content was (2.7 ± 0.05) × 10^−8^ g (m·s·Pa)^−1^. The increase in WVP is due to the fact that anthocyanins contain a large number of hydrophilic groups, which promote the water absorption and expansion of the film, making water molecules more easily pass through the composite film, thereby reducing the barrier efficiency and thus improving the water vapor transmission coefficient [30].

### 3.6. FTIR Analysis of Films

The FTIR spectroscopy of PVA/PTA and PVA films was shown in Figure 3c. It was found that the hydroxyl groups of PVA showed a intense and broad peak at 3365 cm^−1^. The peaks observed at 2941 cm^−1^, 1093 cm^−1^ and 1734 cm^−1^ belong to the tensile vibrations of –CH_2_, C–O and the deformation vibration of C=O on PVA, respectively [27,31,32]. Compared to PVA (a), PVA/PTA (b) film showed new peaks at 1016 cm^−1^ and 1604 cm^−1^, which are due to the tensile vibration of C–O and C=O by anthocyanins [32]. Furthermore, it was found that the characteristic absorption peak of hydroxyl on PVA shifted from 3365 cm^−1^ to 3396 cm^−1^ after combination with PTA, this may be due to the interaction between the hydroxyl groups (–OH) in PTA and PVA through hydrogen bonding, resulting in a shift of the hydroxyl absorption peak. [13,33].

### 3.7. Antioxidant Activity of Films

The antioxidant activity of the films was significantly improved (*p* < 0.05) with increasing anthocyanin concentration, as shown in Figure 3b. The DPPH radical scavenging rates of anthocyanin films for 0%, 1%, 3%, and 5% are 10.94%, 12.55%, 41.89%, 43.46%, respectively. This improvement can be attributed to the polyphenolic nature of anthocyanins, which act as antioxidants by eliminating free radicals through the formation of phenoxy groups [34]. It is important to note that there is a partially significant difference (*p* < 0.05) among the different concentrations. The lower antioxidant rate observed for anthocyanin films at 1% concentration may be due to various factors, such as cross-linking of the active ingredient with the polymer, solubility, and microstructure of the film as well as the release environment [35]. Additionally, it is worth mentioning that crosslinking is often observed to impact properties beyond just antioxidant activity, including mechanical and barrier properties.

### 3.8. Application of PVA/PTA Films for Monitoring Shrimp Freshness

Fresh shrimp freshness was examined in PVA/PTA films with different concentrations of PTA at 4 °C with Figure 4. Enzymes and microorganisms are the main factors in the spoilage of aquatic products, a process that is often accompanied by a rise in pH and the production of volatile compounds such as ammonia and amines during protein degradation, which enhances the fresh shrimp’s off-flavor [34,36]. Nitrogen-containing compounds in alkaline pH can cause changes in the structure of PTA, which can result in color changes. Color changes were observed in PVA/PTA films with different concentrations of PTA. Among them, the film with a 5% PTA concentration showed the most significant visual color change during storage time of 0–4 d; the color of the films shifted roughly as follows: dark purple → bluish-green → light yellow.

As shown in Table 3, during a 48 h storage period, the TVB-N in the shrimp showed a significant increase, rising from an initial (7.2 ± 0.3) mg per 100 g to (28.6 ± 0.01) mg per 100 g. As per the Chinese National Standard (GB 2733-2015) for marine fish and shrimp, TVB-N levels should be kept below 30 mg per 100 g of consumption. This indicates that after 48 h of storage, the shrimp samples had nearly reached the spoilage threshold and were deemed unsuitable for consumption. The initial pH value of fish was 7.1, similar to the variation in TVB-N value, which increased gradually during storage at 4 °C. Notably, the color changes of the chromatic films correlated closely with the shifts in TVB-N and pH values in the shrimp. Throughout the monitoring period, the variations in color of the colorimetric films for 5%-PTA film became increasingly pronounced as the TVB-N levels rose. Initially, the 5% PVA/PTA film was purple, which then transitioned to blue-green at 48 h, and by 92 h, it had changed to a yellow-green hue, with the ΔE values of film changing from 65.94 ± 0.42 to 30 ± 0.1 (as shown in Table 3). The change in total color difference is obvious and can be recognized by the naked eye.

Microbial growth in freshness shrimp during storage and the TPC are shown in Table 3. The acceptable microbiological limit for fresh shrimp was 7 log CFU/g [20]. During the storage, the microorganisms gradually increased incrementally, and the microbial count was about to reach its peak on day 4. As fresh shrimp began to have a slight odor at Day 2, the shrimp were deemed inedible based on odor at Day 4, and all were spoiled at Day 5, consistent with the TVB-N value of fresh shrimp. Due to the significant color change of shrimp during storage, the film developed in this study can be used as a colorimetric index to visualize the freshness of shrimp.

## 4. Conclusions

In the study, the different concentrations of PTA were added into PVA to prepare intelligent packaging films for real-time monitoring of fresh shrimp freshness. It was found that the physicochemical properties of PVA films were improved under the action of PTA with the enhancement of the tensile strength, contact angle, and antioxidant activity of PVA, indicating that PVA/PTA films have the potential to be used as active packaging materials. However, with the addition of PTA, the elongation at break and barrier were significantly decreased (p < 0.05). FTIR results showed a chemical interaction between PTA and PVA. Through Pearson correlation analysis, it is found that some properties of the film are dominantly correlated and have mutual influences. When the films were applied to shrimp freshness detection, visible changes in the film were observed. Therefore, the prepared PVA/PTA films have potential applications in intelligent packaging as a smart label indicator. 

## Figures and Tables

**Figure 1 polymers-16-00495-f001:**
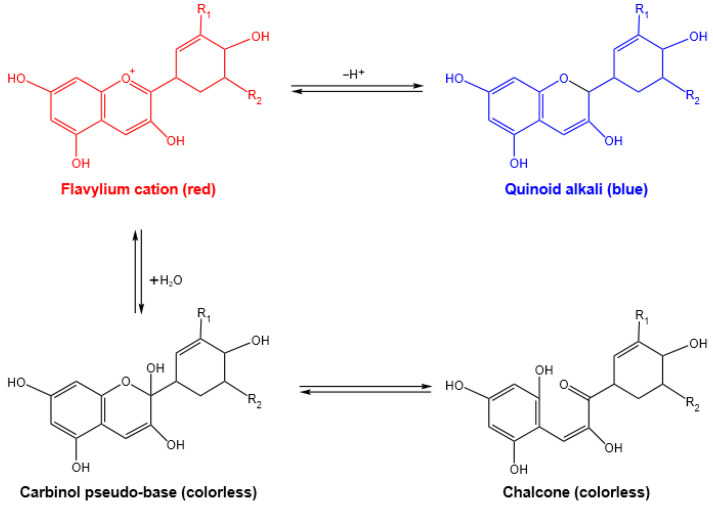
Structural changes of anthocyanins at different pH values.

**Figure 2 polymers-16-00495-f002:**
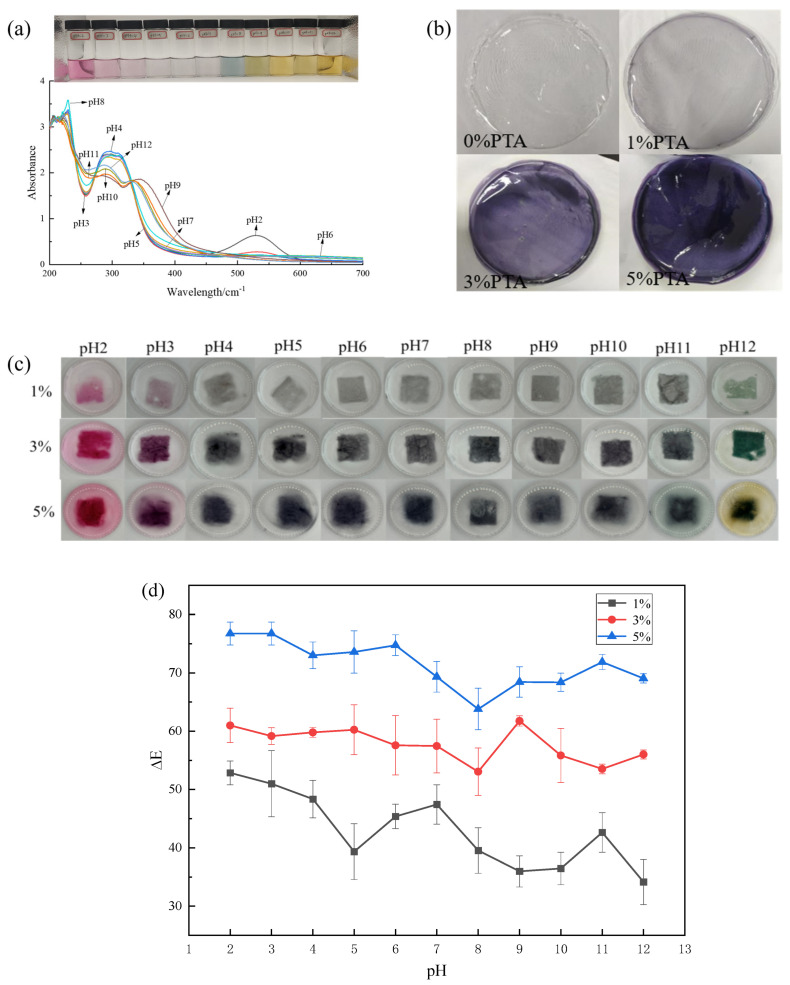
UV-visible spectra (**a**); visual color of PTA at pH 2–12 (**b**); color response of PVA/PTA films with different concentrations of PTA at pH 2–12 (**c**); and total color difference (ΔE) of PVA/PTA films containing 1%, 3%, and 5% PTA, respectively, as a function of pH (**d**).

**Figure 3 polymers-16-00495-f003:**
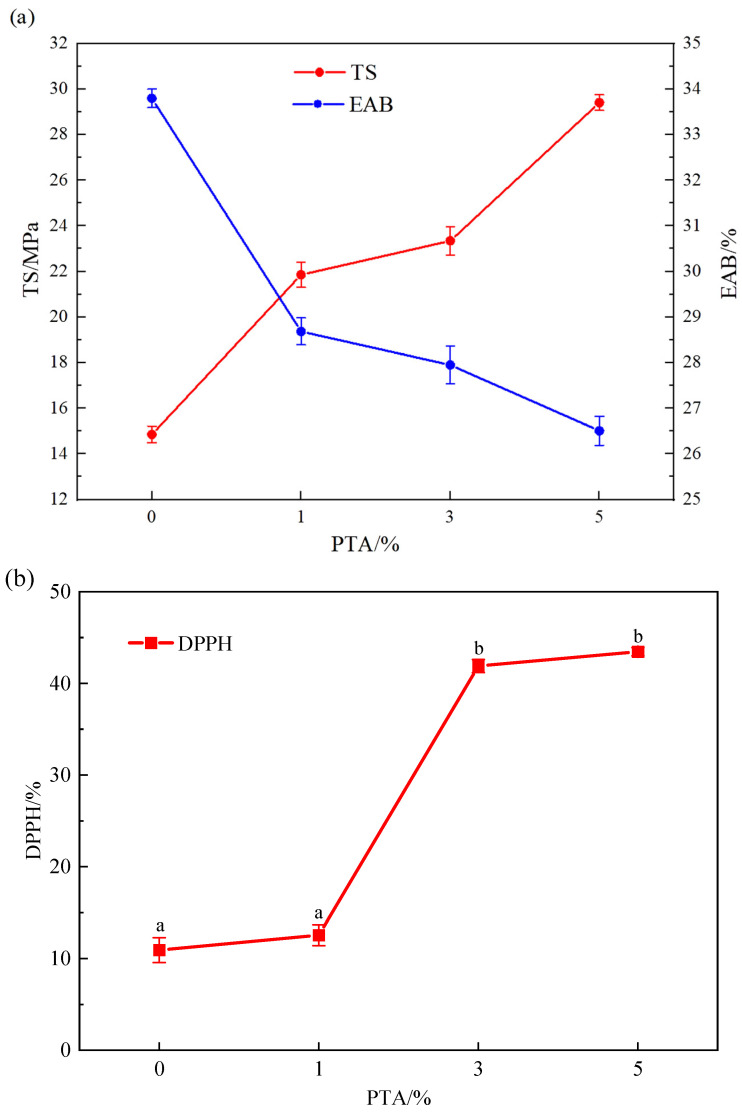
Mechanical properties of the PVA/PTA films (**a**); DPPH radical scavenging activity of PVA/PTA films (**b**), different lowercase letters indicate significant differences (*p* < 0.05); and FTIR spectra of PVA and PVA/PTA films (**c**).

**Figure 4 polymers-16-00495-f004:**
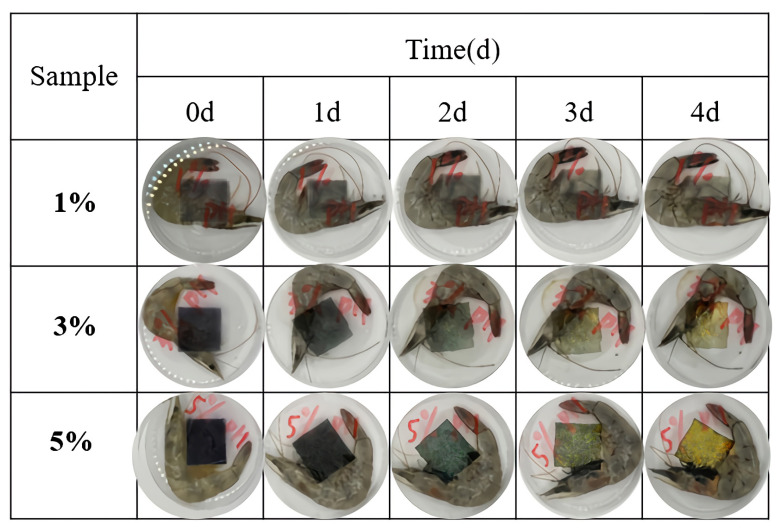
Color change of films based on PVA/PTA films used to monitored shrimp freshness.

**Table 1 polymers-16-00495-t001:** Color parameters (*L**, *a**, *b** and Δ*E**) of original PVA/PTA films.

PTA (%)	*L**	*a**	*b**	Δ*E**
0%	85.16 ± 0.15 ^a^	3.26 ± 0.05 ^c^	−8.8 ± 0.25 ^a^	14.32 ± 0.21 ^d^
1%	57.16 ± 3.64 ^b^	6.32 ± 0.79 ^a^	−9.76 ± 0.43 ^a^	37.21 ± 0.36 ^c^
3%	45.68 ± 3.41 ^c^	5.22 ± 0.25 ^b^	−12.68 ± 1.06 ^b^	48.67 ± 3.01 ^b^
5%	27.06 ± 1.01 ^d^	4.84 ± 0.59 ^b^	−11.7 ± 1.72 ^b^	65.94 ± 0.42 ^a^

Note: Values in the table are expressed as mean ± standard deviation; different lowercase letters indicate significant differences (*p* < 0.05).

**Table 2 polymers-16-00495-t002:** Film thickness, water content, and water vapor transmission coefficient.

PTA (%)	Thickness (μm)	MC (%)	WVP (10^−8^ g Per m. s. Pa)
0%	57.86 ± 2.4 ^c^	6.82 ± 0.26 ^a^	2.08 ± 0.33 ^b^
1%	62.2 ± 8.14 ^b,c^	6.58 ± 0.16 ^a^	1.72 ± 0.23 ^b^
3%	73.48 ± 9.98 ^b^	6.22 ± 0.59 ^a^	1.24 ± 0.02 ^c^
5%	88.22 ± 8.41 ^a^	6.40 ± 0.37 ^a^	2.7 ± 0.05 ^a^

Note: Values in the table are expressed as mean ± standard deviation; different lowercase letters indicate significant differences (*p* < 0.05).

**Table 3 polymers-16-00495-t003:** Changes of freshness monitor (TPC, pH, TVB-N) and Δ*E* values of shrimp using 5% PVA/PTA indicator films during storage at 4 °C.

Time (d)	pH	TVB-N Content (mg Per 100 g)	TPC (log CFU Per g)	Δ*E*
0	7.1 ± 0.2	7.2 ± 0.3	5.261 ± 0.2	65.9 ± 0.4
1	7.4 ± 0.1	19.3 ± 0.1	5.378 ± 0.1	58.78 ± 0.2
2	7.8 ± 0.1	28.6 ± 0.1	5.457 ± 0.1	44.18 ± 0.2
3	8.0 ± 0.1	38.0 ± 0.2	5.659 ± 0.1	35.89 ± 0.2
4	8.4 ± 0.1	50.0 ± 0.1	6.767 ± 0.2	30.12 ± 0.1

Note: Values in the table are expressed as mean ± standard deviation.

## Data Availability

Data are contained within the article.

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
