# Peer review of "Colorimetric Films Based on Polyvinyl Alcohol and Anthocyanins Extracted from Purple Tomato to Monitor the Freshness of Shrimp"

_polymers, 2024, doi:10.3390/polym16040495_

Round 1
Reviewer 1 Report
Comments and Suggestions for Authors
The article "Colorimetric Films Based on Polyvinyl Alcohol and Anthocyanins Extracted from Purple Tomato to Monitor the Freshness of Shrimp" presents an innovative and environmentally friendly approach to food packaging. The integration of anthocyanins extracted from purple tomatoes into polyvinyl alcohol (PVA) to create colorimetric films is a brilliant example of utilizing natural compounds for practical applications. These films, which change color to indicate the freshness of shrimp, offer a smart and intuitive way for consumers to assess food quality.
The study's thorough examination of the films' properties, including their mechanical characteristics, antioxidant activity, and permeability to water vapor, is commendable. The increase in tensile strength and improvement in antioxidant properties with the addition of anthocyanins highlight the dual benefits of enhanced film durability and potential health advantages.
Furthermore, the application of these films in smart packaging represents a significant advancement in food technology. The ability to monitor the freshness of shrimp through color changes addresses a crucial need in food safety and waste reduction. It empowers consumers to make informed decisions about the food they consume while also potentially extending the shelf life of perishable items.
I have some comments to improve the manuscript:
- Materials: add CAS to reagents
- Add in introduction explanation of abbreviation of PTA. All abbreviations should be explained first time in text of manuscript and not only in Abstract.
- Line 74: in 500 ml of solution of what ?
- Line 78: add space before unity of cm. Check whole manuscript.
- Line 85 add space before unity of nm. Check whole manuscript.
- 2.3.8: against which bk was performed dpph?
- Could you include and comment a Pearson correlation matrix encompassing all the study’s results to examine the interrelationships and influences between the various variables?
Reviewer 2 Report
Comments and Suggestions for Authors
The manuscript describes the production of PVA films incorporating PTA extracted from purple tomato, designated color indicators for assessing food freshness. The topic of the paper falls within the scope of the Journal- However, it is regrettable that the manuscript requires additional refinement and several inquiries need to be addressed before it can meet the standards for publication. For instance, the absence of informationregarding the extraction and characterization of PTA from purple tomatoes is a notable gap in the text. Assignation of FTIR bands needs a revision. Please see my suggestions and queries in the PDF file, as comments.

English Grammar and Spelling are Ok.
Round 2
Reviewer 1 Report
Comments and Suggestions for Authors
The manuscript has been significantly improved. No more comments.
Author Response
Dear Reviewer,
Thank you for reviewing my submitted manuscript. I have received your response and noted that you did not provide any comments or suggestions. I greatly appreciate the recognition and approval you have given to my paper.
Once again, thank you for your review and feedback.
Best regards,
Yangyang Qi
Reviewer 2 Report
Comments and Suggestions for Authors
The author has done commendable work, and the manuscript has shown improvement from its initial version. However, there are several issues and inconsistencies that necessitate a thorough revision. For instance, in explaining the migration of PTA, the authors claim that films with 3% and 5% PTA have "unstable interactions," but they also state that the same films exhibit strong interactions when discussing tensile properties. Another inconsistency arises in the analysis of the impact of increasing MC on antioxidant activity (Pearson correlation analysis), as MC was found to be independent of PTA and all films displayed 6% MC. How do the authors explain this? Please refer to the observations outlined in the attached PDF file. Based on my assessment, I recommend major revisions.

Round 3
Reviewer 2 Report
Comments and Suggestions for Authors
I am grateful for the authors' efforts in responding to the queries raised in the second revision of the manuscript. However, there are still certain unresolved issues. Some of the responses, particularly queries 7-9, lack precision. The removal of lines related to interactions raises concerns as the assumption of interactions is not experimentally supported (poor analysis of FTIR). For example, the authors should explain how interactions impact the UV-vis absorption characteristics of films if the purple hue is a result of interactions between PVA and PTA, as stated in the text. Furthermore, the justification of the MC of films needs to be revised, specifically regarding "PTA crosslinking with water molecules", as the authors stated in the text. I recommend conducting experimental research to support your assumptions in this text, rather than leaving it for future work, as mentioned in the answer to the queries. Hence, despite the seemingly minor nature of the revisions, they hold substantial importance, thereby a final round of revision is recommended.

Author Response
请参阅附件。
